Effect of the diet level of whole-plant corn silage on the colonic microflora of Hezuo pigs

Yang Qiaoli 1
Wang Longlong 1
Wang Pengfei 1
Yan Zunqiang 1
Chen Qiong 2
Zhang Pengxia 1
Li Jie 1
Jia Rui 1
Li Yao 1
Yin Xitong 1
Gun Shuangbao 1 3 4 gunsbao056@126.com
1 College of Animal Science and Technology, Gansu Agricultural University , Lanzhou, Gansu , China
2 Gansu Bailu Grass Industry Development Co., Ltd. , Lanzhou, Gansu , China
3 Gansu Diebu Juema Pig Science and Technology Backyard , Diebu , China
4 Gansu Research Center for Swine Production Engineering and Technology , Lanzhou , China
Banaszak Anastazia
Electronic publication date: 2024 Nov 25
Publication date: 2024
Volume: 12
Electronic Location ID: e18630
Received 2024 Jun 11; Accepted 2024 Nov 12
Copyright: © 2024 Yang et al.
Copyright year: 2024
Copyright holder: Yang et al.
License: This is an open access article distributed under the terms of the Creative Commons Attribution License, which permits unrestricted use, distribution, reproduction and adaptation in any medium and for any purpose provided that it is properly attributed. For attribution, the original author(s), title, publication source (PeerJ) and either DOI or URL of the article must be cited.
License URL: https://creativecommons.org/licenses/by/4.0/

Keywords: Hezuo pig, Corn silage, Colon, Intestinal microflora, 16S rRNA

Funding: Natural Science Foundation of Gansu Province 23JRRA1422 Animal Husbandry Pig Industry Technology Innovation Team Project of Gansu Agricultural University GAU-XKTD-2022-25 Natural Science Foundation of Gansu Province 22JR5RA874 Education Science and Technology Innovation Project of Gansu Province GSSYLXM-02 This work was supported by the Natural Science Foundation of Gansu Province (23JRRA1422), the Animal Husbandry Pig Industry Technology Innovation Team Project of Gansu Agricultural University (GAU-XKTD-2022-25), the Natural Science Foundation of Gansu Province (22JR5RA874), and the Education Science and Technology Innovation Project of Gansu Province (GSSYLXM-02). The funders had no role in study design, data collection and analysis, decision to publish, or preparation of the manuscript.

==============================
Background

Whole-plant corn silage (WPCS) is an important roughage source for livestock, and have critical influences on rumen or intestinal microbiota, thus affecting the growth performance and feed efficiency. Our previous studies showed that adding WPCS to the diet of Hezuo pigs could promote the growth and fiber digestibility. While the aim of this study is to understand the effect of dietary WPCS on the colonic microflora in Hezuo pigs, which is essential for improving the roughage exploitation of pigs.

Methods

Thirty-two Hezuo pigs with similar body weight (7.88 ± 0.81) kg were selected and randomly divided into four groups with eight pigs in each group. Pigs in the control group were fed a basal diet, pigs in the experimental groups (Groups I, II, and III) were fed basal diet supplemented with 5%, 10%, and 15% WPCS, respectively, under 120 d experimental period. Six pigs from each group were picked for collecting colonic contents samples. 16S rRNA sequencing was performed to analyze the colonic microbiota of experimental pigs.

Results

The results showed that community richness indexes Chao1 and Observed_species in group III of Hezuo pig were significantly lower than that of the other three groups, community diversity indexes Shannon and Simpson were significantly higher in group I and II in comparison to the control group, and significantly lower in group III in comparison to the control group, group I and II. Adding WPCS to the diet of Hezuo pigs has no influence on the colonic dominant phylum, Clostridium sensu stricto 1 and Rikenellaceae RC9 gut group were most prevalent in the colon of Hezuo pig. When compared with the control group, the relative abundance of Streptococcus was significantly decreased in three experimental groups, while p-251-o5, Parabacteroides, Prevotellaceae UCG-003, Prevotellaceae UCG-001, and F082 exhibited significantly higher relative abundances in at least two experimental groups. Fibrobacter, Rikenellaceae RC9 gut group in group I, UCG-010 in group II, Bacteroides in group III exhibited increased relative abundance as compared with the control group. PICRUSt functional annotation indicated that the functions of cellular process and signaling were significantly increased in all WPCS-rationed groups, cancers, nervous system, immune system and environmental adaptation were all differed from groups I and II; three predominant pathways of translation, nucleotide metabolism and signal were only differed from the group II.

Conclusions

Feeding with 5% and 10% WPCS for Hezuo pigs could improve their colonic microflora diversity, and increase the relative abundance of fiber-digesting bacteria, which may potentially help to improve the fibre digestibility of Hezuo pigs by regulating the microbial function of cellular process and signaling, nucleotide metabolism, translation.

Introduction

In animal feeding, roughage is a crucial ingredient. For a long time, corn stover has been a principal roughage resource for livestock. Compared with corn stover, whole-plant corn silage (WPCS) represents a superior choice for livestock nutrition, attributed to its appealing taste, tender consistency, and distinctive acidic aroma (Cui et al., 2022). Additionally, the nutritional value of WPCS is similar to that of the concentrate feed because of containing corn kernels (Moloney & Drennan, 2013). As a basal diet for ruminant animals, WPCS has been reported to significantly improve their production performance and feed utilization efficiency (Zaralis et al., 2014; Cui et al., 2022). What’s more, most of these studies have focused on the contribution of the rumen microflora in helping ruminants digest and utilize WPCS (Guo et al., 2022). Cui et al. (2022) found that feeding WPCS could improve the level of rumen fermentation and growth performance of crossbred Simmental cattle, mechanically, which attributed to alter the composition of rumen microbiota and further facilitate the intestinal nutrient metabolism of amino acids, carbohydrates, and nucleotides. Guo et al. (2022) found that whole-plant corn silage inoculated with bacterial inoculant can improve the fibre digestibility in sheep by increasing the relative abundance of rumen microflora Prevotella and Bacteroidetes, and/or by recruiting some functions involved in glycolysis/gluconeogenesis and citrate cycle pathways. The above research indicates that gastrointestinal microbes play a critical role in the digestion and utilization of WPCS in ruminants.

Feed costs represent a significant proportion of the total expenditure incurred in pig production, estimated to be 60% to 70% of total costs. Consequently, there is an urgent need to identify strategies to reduce feed costs and improve feeding efficiency in commercial pig production. Compared to refined feed, roughage is usually less expensive. Increasing in the proportion of roughage in pig diet is one of the main ways of effectively reducing feed costs (Woyengo, Beltranena & Zijlstra, 2014). However, straw-based roughage is known to have higher crude fibre content and lower nutrient availability. It can regulate the composition of gut microbiota by adding an appropriate level of dietary fibre (Heinritz et al., 2016; Zhang et al., 2020; Pi et al., 2021), thus facilitating the gastrointestinal peristalsis and improving the intestinal development and physiological functions of animal’s (Koh et al., 2016). As a monogastric omnivore, pigs have a well-developed cecum and colon, which contain various of microorganisms essential for fermenting crude fibre and utilizing organic matter (Wang et al., 2024). Nevertheless, several studies have reported that the use of WPCS in pig production have effects on the development and health of stomach, nutrient utilization, growth and carcass performance of fattening pigs (Mason et al., 2013; Zanfi et al., 2014), it remains unclear how the colonic microflora respond to the nutrient utilization of WPCS rations in pigs.

Hezuo pig is one of a breed of Tibetan pigs that have been semi-grazed in the alpine pastures of Gannan Tibetan Autonomous Prefecture, Gansu Province (Tang et al., 2023). They are known for their better meat quality, strong abilities of disease resistance, hypoxic tolerance and digestion the non-conventional high-fiber feed, such as dregs and grass products (Chang et al., 2022). In particular, it has significant advantages in utilizing a wide range of roughage resources (Niu et al., 2022a; Wang et al., 2024). Our previous study found that the addition of WPCS can improve the feed conversion rate of Hezuo pigs (Yin et al., 2024). Based on these foundations, the present study further investigates the effects of different levels of WPCS on the composition and function of the colonic microflora in Hezuo pigs, which will provide crucial foundation for further exploring the microbiota that may be related to the roughage tolerance and analyzing the contributions of the intestinal microflora in utilizing roughage in Hezuo pigs.

Materials and Methods

Experimental design

WPCS was provided by a dairy plant in Yongdeng Farm (Gansu Province, China). The WPCS was naturally air-dried and crushed into powder to prepare experimental diets, and the traditional nutritional levels of WPCS are shown in Table 1.

Table 1 The general nutrient contents of whole plant corn silage (dry matter based).

Nutrient	Content (%)	
CP	11.36	
EE	4.66	
NDF	41.29	
ADF	29.29	
Ash	7.49	
Ca	2.47	
P	0.32	
Note:

CP, crude protein; EE, ether extract; NDF, neutral detergent fibre; ADF, acid detergent fibre.

The experimental rations were formulated according to the China feeding standard of swine (NY/T 65-2004, https://www.antpedia.com/standard/5041407.html), the crushed WPCS powder was added to replace the basal diet at the proportions of 5%, 10% and 15% to total weight, and ensured that meet the pig energy requirements. Once the test diets had been prepared and mixed uniformly, they were placed in ration pouches with inner bags to prevent moisture ingress. The dietary ingredients of experimental pigs and the nutrient contents of experimental diets are shown in Tables 2 and 3, respectively.

Table 2 Dietary ingredients of experimental pigs.

Ingredients (%)	Diets	
Control	Group I (5% WPCS)	Group II (10% WPCS)	Group III (15% WPCS)	
Corn	69	66.1	63.2	60.3	
Soybean meal	20	19.9	19.8	19.7	
4% Premix	4	4	4	4	
Bran	7	5	3	1	
Whole plant corn silage	0	5	10	15	
Note:

Premix provided the following per kg diets: Fe (as ferrous sulfate) 2∼7 g; Cu (as cupric oxide) 0.2∼0.625 g; Zn (as zinc sulphate) 1.0∼2.0 g; Mn (as manganese sulfate) 0.5∼1.5g; Se (as sodium sulfate) 3.0∼10.0 mg; I 3.0∼25.0 mg; VA100∼160 KIU; VD 30∼80 KIU; VE 290 mg; VK 30.0∼90.0 mg; VB1 25 mg; VB6 25 mg; VB12 0.2 mg; Pantothenic acid 180 mg; Nicotinic acid 240 mg; Folic acid 10 mg.

Table 3 The nutrient contents of experimental diets.

Nutrient contents	Diets	
Control	Group I (5% WPCS)	Group II (10% WPCS)	Group III (15% WPCS)	
DE (MJ/kg)	13.47	13.35	13.23	13.10	
CP (%)	15.89	15.74	15.60	15.46	
CF (%)	3.45	4.65	5.85	7.05	
Ca (%)	0.50	0.62	0.74	0.85	
P (%)	0.35	0.35	0.35	0.36	
NDF (%)	11.80	12.83	13.87	14.91	
ADF (%)	5.25	6.34	7.43	8.53	
Note:

The values of nutrient contents were calculated based on dry matter. DE, digestive energy; CP, crude protein; CF, crude fibre; Ca, calcium; P, phosphorus; NDF, neutral detergent fibre; ADF, acid detergent fibre.

Forty healthy Hezuo pigs at 60 days of age with similar body weight were selected and fed with the basal diet in the pre-feeding period for 7 days. Then, 32 Hezuo pigs, which had no differ in terms of feed intake and weight gain (body weight 7.88 ± 0.81 kg), were selected and divided equally into four treatment groups, with eight replicates per group (with half of male and half of female) and one pig per replicate. During the main experimental period, the control group was fed the basal diet, and the experimental groups I, II, and III were fed experimental diets that was replaced with 5%, 10%, and 15% WPCS in basal diet, respectively, for 120 days. The feed amount was based on 3% of live weight, and weighed before and after of each feeding to calculated the feed consumption of each pig. All pigs were raised in the same house and kept under the same housing and management conditions, and each replicate was housed in one single pen for the duration of the experiment. They were fed three times a day at 08:00, 13:00, and 19:00, and were given ad libitum access to water. Vaccinations, deworming, cleaning and disinfection are in line with routine pig farm practice. The average of ambient temperature and relative humidity of the pig housing were recorded as (24 ± 2) °C and 60% to 65%, respectively.

The experiment was conducted from June to October 2022 at the Gansu Bailu Eco-Tourism Farm (Lanzhou, China). All experimental protocols used in this study were reviewed and approved by the Animal Protection and Use Committee of Gansu Agricultural University (approval number: GSAU-Eth-AST 2022-024).

Sample collection and sequencing procedures

At the end of the experiment, six pigs in each group were randomly selected for sacrifice after barbiturate anesthesia. The intestines were separated and the contents samples in colonic midpiece were collected in 5 mL sterilized centrifuge tubes, which were immediately put into liquid nitrogen quick-freezing, and stored at −80 °C.

Illumina sequencing of the 16S rRNA gene was carried out to characterize the diversity and composition of the microbial community.

Sequencing analysis of colonic content samples for bacterial flora

A TIANGEN DP302-02 bacterial genomic DNA extraction kit (TIANGEN, Beijing, China) was used to extract total bacterial DNA, following the instructions provided. The extracted total DNA was purified and its purity and concentration were determined by 2% agarose gel electrophoresis, and the samples were diluted to 1 ng/μL with sterile water. Diluted bacterial genomic DNA was used as a template for amplification the V4 hypervariable region of the 16S rRNA gene using the primers (515F: 5′-GTGCCAGCMGCCGCGGGTAA-3′ and 806R: 5′-GGACTACHVGGGTWTCTAAT-3′). The library was constructed using the NEB Next® Ultra™ II FS DNA PCR-free Library Prep Kit (New England Biolabs, Ipswich, MA, USA) following the manufacturer’s recommendations. The constructed library was quantified by Qubit and qPCR. After library qualification, the NovaSeq 6000 was used for on-line sequencing of PE250 at Novogene Bioinformatics Technology Co.

Bioinformatics and statistical analysis

The acquired reads were merged using FLASH (Version 1.2.11, http://ccb.jhu.edu/software/FLASH/) (Magoc & Salzberg, 2011) and quality filtering of raw reads was strictly filtered by FASTP software (Version 0.23.1) to obtain high quality clean tags following the QIIME (V1.9.1) quality control procedure (Bokulich et al., 2013; Caporaso et al., 2010). Subsequently, the clean tags were compared with the species annotation database (Silva database, https://www.arb-silva.de/) to detect chimeric sequences. The chimeric sequences were then removed to obtain the final effective tags (Edgar et al., 2011).

The Uparse algorithm (Uparse v7.0.1001) is used to cluster all effective tags (Edgar, 2013), and the sequences were clustered into OTUs (Operational Taxonomic Units) with a 97% identity threshold. The most abundant sequences in the OTUs were screened as representative sequences for annotating species (Silva 138.1 database, https://www.arb-silva.de/documentation/release-1381/) (Quast et al., 2013). All raw sequences of the 16S rRNA gene have been deposited in the Sequence Reads Archive (SRA) under the BioProject accession number PRJNA1071212.

Venn diagrams are created in using the VennDiagram function in R software (version 4.0.3). The community alpha diversity was calculated using Qiime2 (Version 2023.5). Principal coordinates analysis (PCoA) analyses were calculated and plotted using the ade4 and ggplot2 packages in R software package (version 4.0.3; R Core Team, 2020). Based on the species annotation results, the top 10 species in terms of relative abundance were selected to plot the histogram of species abundance for each group at phylum and genus levels using the SVG function in Perl. The T-test in R was used to analyze the significant differences of species. LEfSe analysis was performed using Lefse software with the default setting of LDA score threshold of 4. The function of microflora was predicted using PICRUSt2 (V2.3.0) software.

The statistical differences in alpha diversity of bacterial communities were detected using Tukey test. Differentially abundant bacterial taxa and bacterial function between the control group and the three experimental groups were detected by T-test. The differential analyses of bacterial genera of control vs group I, control vs group II, and control vs group III was conducted using the DESeq2 package in RStudio (version 1.20.0; R Studio Team, 2020). Genera with padj ≤ 0.05 and |log2FoldChange| ≥ 1 were classified as either upregulated or downregulated across the various dietary groups.

Results

Effect of whole-plant corn silage on the growth performance of Hezuo pigs

In our prior research, we evaluated the impact of varying levels of whole-plant corn silage incorporation on the growth performance metrics, including body weight, average daily feed intake, and feed conversion ratio in Hezuo pigs (Yin et al., 2024; Table 4), and found that the feed conversion ratio in the 10% supplementation group (Group II) was substantially lower than that of the control group (P < 0.05), while the other groups exhibited no substantial differences (P > 0.05). The average daily feed intake and average daily weight gain showed no significant variation across the four groups (P > 0.05).

Table 4 Effect of whole-plant corn silage on the growth performance of Hezuo pigs.

Performance metrics	Diets	
Control	Group I (5% WPCS)	Group II (10% WPCS)	Group III (15% WPCS)	
Initial body weight/kg	8.06 ± 0.25	7.78 ± 0.37	7.56 ± 0.40	8.12 ± 0.18	
Final body weight/kg	39.60 ± 1.17	40.81 ± 2.74	43.36 ± 1.93	39.62 ± 2.07	
Average daily feed intake/kg	1.08 ± 0.04	1.10 ± 0.06	1.16 ± 0.05	1.07 ± 0.05	
Average daily weight gain/kg	0.26 ± 0.01	0.28 ± 0.02	0.30 ± 0.02	0.26 ± 0.02	
Feed conversion ratio	4.11 ± 0.09a	4.01 ± 0.07ab	3.89 ± 0.03b	4.10 ± 0.08ab	
Note:

Within the same row, values that possess distinct superscript letters exhibit significant differences (P < 0.05). WPCS, whole-plant corn silage. These data were previously published in our study conducted by Yin et al. (2024).

Venn diagram of the colonic microflora in Hezuo pigs

As shown in Fig. 1, a total of 5,655 OTUs were obtained from the colonic content samples of all experimental pigs. Of these, 979 OTUs were common to the four groups, while 771, 691, 1,575, and 471 OTUs were specific to the control group, group I, group II, and group III of experimental pigs, respectively. Notably, the pigs in experimental II exhibited the highest species richness of colonic flora among the four groups.

Figure 1 Venn diagram of colonic flora in Hezuo pigs.

Alpha diversity of the colonic microflora in Hezuo pigs

Figure 2 demonstrated that there was no significant difference for the Chao1 and Observed_species indexes among the control, group I, and group II (P > 0.05), while these indexes were all significantly higher than those observed in group III (P < 0.05, Figs. 2A and 2B). The Shannon and Simpson indexes were found to be significantly higher in group I and II in comparison to the control group, and significantly lower in group III in comparison to the control group, group I and group II (P < 0.05, Figs. 2C and 2D).

Figure 2 Microbial α-diversity indices of the colonic intestinal tract of Hezuo pigs.

The same letter indicates a non-significant difference (P > 0.05) and different letters indicate a significant difference (P < 0.05).

PCoA analysis of the colonic microflora in Hezuo pigs

As illustrated in Fig. 3, principal components PC1 and PC2 accounted for 15.22% and 13% of the total variables, respectively. This indicates good clustering of samples in each group. The colonic microflora of the control group and test group I Hezuo pigs were more closely clustered, and the samples of test group II and test group III were more closely clustered. What’s more, the colonic microflora in control and group I of Hezuo pigs makes sharp distinctions to the groups II and III, indicating that adding 10% and 15% WPCS in the ration of Hezuo pigs have resulted in significant changes of the structure and diversity of the colonic microflora.

Figure 3 Weighted UniFrac distance PCoA based analysis.

Composition of the colonic microflora in Hezuo pigs

Figure 4 presents the most abundant top 10 species at the phylum and genus level for each group of Hezuo pig. The predominant bacterial phyla in the colonic microflora of Hezuo pigs were Firmicutes, Bacteroidota, and Euryarchaeota, collectively accounting for over 95% of the colonic microflora in both the control and group II. The remaining bacterial phyla were Spirochaetota, Proteobacteria, Chloroflexi, Fibrobacterota, Desulfobacteroa, Verrucomicrobiota, and Actinobacteriota (Fig. 4A). As shown in Fig. 4B, the most prevalent bacterial genera in the colon of Hezuo pig were Clostridium sensu stricto 1 and Rikenellaceae RC9 gut group with the highest abundance observed in the four groups. Additionally, Streptococcus was more prevalent in the control group, p-251-o5 was more common in the three experiment groups. The remaining major bacterial genera were Parabacteroides, Methanobrevibacter, Bacteroides, Lactobacillus, and Prevotella.

Figure 4 Top 10 bacteria at the phylum level (A) and genus level (B) in the colonic of Hezuo pigs.

Analysis of the differences of the colonic microflora in Hezuo pigs

The differences of the colonic bacterial genera in Hezuo pigs between the control group and the three experimental groups are presented using a volcano plot and T test bar plot, respectively (Figs. 5 and 6). In comparison to the control group, there were 14, and five upregulated genera, five, and 10 downregulated genera in groups I, and II, respectively (Figs. 5A and 5B). The group III is the most exclusive compartment, enriching for 20 genera while downregulating 24 genera (Fig. 6C). Relative abundance of Streptococcus was significantly decreased in experimental groups I, II, and III (P < 0.05, Fig. 6). Relative abundances of Lachnospira, Eubacterium_xylanophilum_group and Eubacterium_siraeum_group in experimental groups II and III were observed significantly decreased when comparing with the control group (P < 0.05, Fig. 6). Relative abundances of p-251-o5, Parabacteroides, Prevotellaceae UCG-003, Prevotellaceae UCG-001, and F082 were higher in at least two experimental groups than that in the control group (P < 0.05, Fig. 6). Furthermore, Fibrobacter, Rikenellaceae RC9 gut group in group I, and UCG-010 in group II, as well as the Bacteroides in group III exhibited a significant increased relative abundance as compared with the control group (P < 0.05, Fig. 6).

Figure 5 (A–C) Volcano blot of differences in colonic microbiota between control and the various dietary groups of Hezuo pigs.

Each point in the diagram signifies a distinct species, with “Up” indicating a species that exhibits greater abundance in the first comparison group compared to the second, while “Down” denotes the contrary. The names of the top ten species were presented based on their significance ranking.

Figure 6 Bar blot of differences in colonic microbiota between control and the various dietary groups of Hezuo pigs.

LEfSe analysis of the differential colonic microflora in Hezuo pigs

The differential colonic microbes among the four groups Hezuo pigs using LEfSe analysis with the standard of LDA score > 4 was shown in Fig. 7. It can be seen that Streptococcus and Lactobacillus were the differential microbial taxa in the control group Hezuo pigs, Prevotellaceae, Lachnospiraceae, Spirochaetales, Treponema and Fibrobacter (and Fibrobacter intestinalis) were the differential microbes in the colon of Group I Hezuo pigs. The differential microflora in the colon of Group II Hezuo pig were Oscillospiraceae, Rikenellaceae_RC9_gut_group, Tannerellaceae, UCG_005 and Christensenellaceae. The differential microflora in the colon of Group III Hezuo pig were p_251_o5, Fibrobacter and Bacteroides.

Figure 7 Species differences in LEfSe analyses of colonic microflora in Hezuo pigs.

Microbial functions in the colon of Hezuo pigs

The top 35 microbial functional categories ranked by their relative abundance, were selected for comparison their differences between the control group and the three experimental groups at the secondary functional level of KEGG pathways. The results are shown in Fig. 8. When compared to the control group, cellular process and signaling were significantly increased, and infectious diseases were significantly decreased in all there comparation groups (P < 0.05); cancers, nervous system, immune system and environmental adaptation were all differed from groups I and II (P < 0.05); xenobiotics biodegradation and metabolism was only differed from the group I (P < 0.05). Notably, three predominant pathways of translation, nucleotide metabolism and signal were only differed from the group II (P < 0.05).

Figure 8 Heatmap of predicted clustering of PICRUSt genes for the colonic microbes of Hezuo pigs.

Discussion

The microbial composition in the intestine of pigs varies with the proportion of dietary fibre (Li et al., 2021), which in turn affects fibre-digesting enzyme activity and short-chain fatty acid concentration. Studies have shown that adding a certain amount of fibre to the pig’s diet, which not only improves performance, but also promotes gastrointestinal development and health (Zhao et al., 2023; Jha & Berrocoso, 2015; Hermes et al., 2009). Due to a lack of endogenous fiber digestion enzymes, the digestion of dietary fibre in pigs relies mainly on microbial fermentation in large intestine (Tardiolo et al., 2023; Sutera et al., 2023). There is therefore the need to understand how the microbial diversity and composition in the large intestine of pigs responds to the roughage levels, which is essential for the roughage exploitation of pigs.

In our previous articles, we described the effects basic dietary inclusions of WPCS on growth performance and fiber digestive characteristics in Hezuo pigs (Yin et al., 2024; Wang et al., 2024), while the aim of this study is to present the effect of dietary WPCS on the colonic microflora in Hezuo pigs.

Alpha diversity reflects the richness and diversity of microbial community (Jiang et al., 2022). In present study, we found that the α-diversity indexes Simpson and Shannon were significantly higher in group I and II in comparison to the control group, while these indexes in group III were significantly lower than other three groups. PCoA analysis showed that the colonic microflora in control and group I makes sharp distinctions to the groups II and III in Hezuo pigs. These results indicate that the addition of 10% WPCS to the diet of Hezuo pigs resulted in a significant increase in the composition and diversity of colonic microflora, whereas 15% WPCS in the diet of Hezuo pigs caused a disruption in the balance of colonic microflora. The findings from the differential genera analysis of the volcano plot in this study corroborate this explanation. Wu et al. (2022) demonstrated that 17% and 24% broad bean straw silage could increase the relative abundance of Spiroplasma, Fibroplasma, and Prevotella in the cecum of Duroc × Bame crossbred pigs. In this study, analysis of the colonic microflora composition of Hezuo pigs revealed that Firmicutes and Bacteroidota were the most dominant bacterial phyla, accounting for over 90% of the total bacteria. Our results concur with those of a previous study in Durco × Bamei crossbred pigs, Firmicutes and Bacteroidota occupied 85% and 3.6% of the control group, respectively (Wu et al., 2022). While in Durco × Bamei crossbred pigs, the relative abundance of Firmicutes was decreased and the relative abundance of Bacteroidota was increased in the experiment group of feed broad bean straw silage. Greenhill (2015) and Shoaie et al. (2013) showed that Firmicutes and Bacteroidota were the most common bacterial phyla in fattening pigs. A variety of cellulolytic bacteria are included in Firmicutes and Bacteroidetes, and most of which can ferment polysaccharides and produce SCFAs (Brulc et al., 2009; Spence, Wells & Smith, 2006). This leads to increased utilization of carbohydrates, proteins, and other substances by the host. The balance between Firmicutes and Bacteroidota is crucial for maintaining the stability and health of the host’s intestinal microecology (Ley et al., 2008; Hooper, 2004; Hooper et al., 2001). For bacterial genera, the results in present study indicate that adding WPCS to the diet of Hezuo pigs significantly increased the relative abundance of the genera p-251-o5, Prevotellaceae UCG-003, Prevotellaceae UCG-001, Bacteroides, Parabacteroides, and Fibrobacter. Studies have reported that Bacteroides and Prevotellaceae UCG-003 could produce fiber-degrading enzymes, including mannanase, xylanase, and β-glucanase, which break down dietary fibers such as cellulose and cellobiose (Flint & Bayer, 2008). Additionally, as cellulolytic bacteria, Fibrobacter and Prevotellaceae UCG-003 is mainly present in the gastrointestinal tract of herbivores and is responsible for degrading indigestible cellulose (Ransom-Jones et al., 2012; Neumann & Suen, 2018). Hooper et al. (2001) reported that the abundance of UCG bacterial family increased significantly with the increase of dietary fiber levels in the caecum of calves. Prevotella_1 and Prevotella_2 have been recognized as promising biomarkers linked to the digestibility of apparent neutral and acid detergent fiber in Suhuai pigs (Niu et al., 2022b). Cui et al. (2022) found that the relative abundance of Prevotella_1 was significantly increased in cattle fed WPCS. Qiu et al. (2022) founded that dietary fiber accelerated the colonization of Rikenellaceae RC9 gut group, Faecalibacterium, and Prevotellaceae UCG 001. Interestingly, the LEfSe analysis in present study showed that Prevotellaceae, Fibrobacter (and Fibrobacter intestinalis) and Rikenellaceae RC9 gut group were the particular bacterial community in the colon of group I and II Hezuo pigs. It has been documented that Fibrobacter_intestinalis, and Prevotellaceae are cellulolytic bacterial species exhibiting significantly high levels of activity within swine intestines (Pu et al., 2022; Metzler & Mosenthin, 2008). This suggests that adding WPCS to the diet of Hezuo pigs stimulated the enrichment of several fiber-degrading bacteria in the colon, which in turn provides the impetus for digesting dietary fiber in Hezuo pigs. Yang et al. (2017) found that Prevotella and Streptococcus exhibited a strong negative correlation in the intestinal flora of healthy piglets. The relative abundance of Streptococcus was significantly lowered in all experimental groups in this study. Pu et al. (2022) reached a comparable conclusion, noting that an increase in fiber intake led to a reduction in the prevalence of Streptococcus in Suhuai pigs. This was probably due to the increase of fiber degradation-related microbial community, which suppresses the colonization and growth of their competitors.

The PICRUSt analysis of the enrichment of bacterial functional pathways between the control group and the experimental groups of Hezuo pigs gave similar results to the afore-mentioned analyses. The striking finding we observed was that the groups received WPCS showed increased relative abundances for the cellular process and signaling, and decreased relative abundance for infectious diseases. It may be as a consequence of the lower relative abundances of Streptococcus and Lactobacillus in experimental groups, as well as the increases of their competitors for cellulolytic bacteria. Streptococcus and Lactobacillus protects the immune function and health of the gastrointestinal tract in animals (Zhang et al., 2015). Three predominant pathways of translation, nucleotide metabolism and signal were only differed from the group II, immune system and environmental adaptation were all differed from groups I and II, which has also been observed in beef cattle and in broilers (Cui et al., 2022; Qiu et al., 2022). Qiu et al. (2022) found that the dietary fiber accelerated the colonization of several cellulolytic bacteria, which further altered the relative abundance of microbial function of carbohydrate metabolism and genetic information processing. Prevotellaceae_UCG-003 in beef cattle with feeding WPCS diet was positively correlated with xenobiotics biodegradation and metabolism, genetic information processing, nucleotide metabolism, and translation (Cui et al., 2022). Microbiota function in host metabolism mainly through carbohydrates fermentation and epithelial cells produce endogenous products that participate in metabolic and immunologic processes (Guarner & Malagelada, 2003). These observations suggest that adding WPCS to the diet of Hezuo pigs can promote the growth of cellulolytic bacteria and improve the metabolic motility of colonic microorganisms, which functions enriched in cellular process and signaling, nucleotide metabolism, and other related functions.

Conclusions

Taken together, feeding with 5% and 10% WPCS for Hezuo pigs could improve their colonic microflora diversity, and increase in the relative abundance of Rikenellaceae RC9 gut group, Prevotellaceae UCG-003, Bacteroides, and Fibrobacter in the present experiment condition. These changes may contribute to improve the fibers digestion in Hezuo pigs by regulating the microbial function of cellular process and signaling, xenobiotics biodegradation and metabolism, nucleotide metabolism, translation.

Supplemental Information

Supplemental Information 1 Author Checklist.

Additional Information and Declarations

Competing Interests

Author Contributions

Animal Ethics

Data Availability

The authors declare that they have no competing interests.

Qiaoli Yang conceived and designed the experiments, performed the experiments, analyzed the data, prepared figures and/or tables, authored or reviewed drafts of the article, and approved the final draft.

Longlong Wang conceived and designed the experiments, performed the experiments, analyzed the data, prepared figures and/or tables, authored or reviewed drafts of the article, and approved the final draft.

Pengfei Wang conceived and designed the experiments, analyzed the data, prepared figures and/or tables, and approved the final draft.

Zunqiang Yan performed the experiments, analyzed the data, prepared figures and/or tables, authored or reviewed drafts of the article, and approved the final draft.

Qiong Chen conceived and designed the experiments, performed the experiments, prepared figures and/or tables, authored or reviewed drafts of the article, and approved the final draft.

Pengxia Zhang performed the experiments, prepared figures and/or tables, and approved the final draft.

Jie Li performed the experiments, analyzed the data, prepared figures and/or tables, authored or reviewed drafts of the article, and approved the final draft.

Rui Jia performed the experiments, analyzed the data, prepared figures and/or tables, and approved the final draft.

Yao Li performed the experiments, authored or reviewed drafts of the article, and approved the final draft.

Xitong Yin conceived and designed the experiments, performed the experiments, prepared figures and/or tables, and approved the final draft.

Shuangbao Gun conceived and designed the experiments, prepared figures and/or tables, authored or reviewed drafts of the article, and approved the final draft.

The following information was supplied relating to ethical approvals (i.e., approving body and any reference numbers):

Animal Protection and Use Committee of Gansu Agricultural University (approval number: GSAU-Eth-AST 2022-024).

The following information was supplied regarding data availability:

The data presented in the study are available at NCBI: PRJNA1071212.

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
