# Peer review of "Effect of the diet level of whole-plant corn silage on the colonic microflora of Hezuo pigs"

_PeerJ, doi:10.7717/peerj.18630_

## Round 0.1 · original submission · Major Revisions

We hvae concluded the evaluation of your submission. Two expert reviewers have evaluated your manuscript and their comments can be seen below and in an attachee PDF. Both comment that the manuscript has merit, but it also has a number of weaknesses throughout the manuscript. In order for us to consider this manuscript further these issues mush be attended to in a revised version of the manuscript. Please ensure that you respond to each issue point by point in a rebuttal, making sure to clearly indicate within the manuscript where any modifications have been made.

Please ensure you thoroughly revise the methods section. For example, the 5%, 10% and 15% WPCS was determined as w:w or vol:vol or some other metric? There are many other details that are missing as pointed out by both reviewers. All other sections of the manuscript also need a complete revision. Fo example, in your conclusions what do you mean by "appropriate ratios". Could you be more specific?

Also make sure that the next version that you upload has been thoroughly reviewed for proper English language use.

Reviewer 1 ·

Basic reporting

The entire paper requires deep revisions before potential publication.
Hereafter, I propose some comments and suggestions for the authors.

The entire abstract text should undergo review for spelling errors and comprehensive editing throughout the manuscript. The introduction could benefit from enhanced exposition.

At line 71, a reference should be added. References are also needed from lines 73 to 76, as well as from lines 87 to 89 and from 95 to 103, specifically on swine breed.

The Materials and Methods section requires improvement with additional details on animal feeding and management. References should be included for methodologies and software used within this section. Footnotes detailing information provided in tables (e.g., Table 1) are missing, including abbreviations.

Was the diet administered based on 3% of live weight? Did the animals consume their entire daily food ration? Were they individually fed in single boxes to accurately assess food consumption?
Details on the dietary aspect should be integrated into the text.

Results on growth performance (e.g., body weight) and indices such as food intake, feed conversion ratio, and average daily gain are absent and should be included in the results.

Regarding QIIME2, an updated software version should be used, as the 2020 version is outdated. Additionally, specify the reference database used to generate OTUs. Silva 138? RDP? Please specify.

It may be beneficial to repeat the bioinformatics analysis using the latest QIIME2 and reference database for OTU classification. Compare the new OTU table with previously obtained results to assess differences. Also, specify the version of Picrust used, ideally the most recent one.

Which R packages were used to estimate alpha and beta diversity? Phyloseq and Vegan packages, respectively? The specific statistical tests employed have not been specified. Please rewrite the paragraph on ‘Data statistics’ to include an explanation of alpha and beta diversity estimation.

The entire Materials and Methods section requires rewriting and updating with appropriate bibliographic references, structured as follows:

Experimental Design:
Include in a single sub-paragraph sections on experimental materials, experimental rations, time and location of the experiment, and experimental design and feeding management.

Sample Collection and Sequencing Procedures:
Include in a single sub-paragraph sections on sample collection and laboratory methods for sample processing and sequencing.

Bioinformatics and Statistical Analysis:
Include in a single sub-paragraph methods for bioinformatics and statistical analysis.

Change ‘Observed_otus’ to ‘Observed_species’ (Observed species index).

Line 206: Instead of listing all metabolic pathways, consider creating a table per group to better outline the results.

In Figure 2, indicate ‘Observed species’ and various indices in uppercase (Shannon, Simpson, etc.). In Figure 4, specify if relative abundance is expressed as a percentage (%).

Italicize all bacterial strain names.

Update the bibliography with recent references.

Authors are encouraged to include the following studies in their manuscript, updating the bibliography with research on alternative feed for pigs:

Tardiolo, G., Romeo, O., Zumbo, A., Di Marsico, M., Sutera, A. M., Cigliano, R. A., Paytuví, A., & D’Alessandro, E. (2023). Characterization of the Nero Siciliano Pig Fecal Microbiota after a Liquid Whey-Supplemented Diet. Animals: an open access journal from MDPI, 13(4), 642. https://doi.org/10.3390/ani13040642

Sutera, A. M., Arfuso, F., Tardiolo, G., Riggio, V., Fazio, F., Aiese Cigliano, R., Paytuví, A., Piccione, G., & Zumbo, A. (2023). Effect of a Co-Feed Liquid Whey-Integrated Diet on Crossbred Pigs’ Fecal Microbiota. Animals: an open access journal from MDPI, 13(11), 1750. https://doi.org/10.3390/ani13111750

Furthermore, another critical aspect overlooked pertains to the raw sequences of the swine metagenome. Have these sequences been deposited in the Sequence Reads Archive (SRA)? Have the authors completed this deposition? If not yet deposited, I urge the authors to submit their sequences and include the Bioproject accession codes in the manuscript. Depositing metagenomic sequences from our studies is crucial for scientific community adherence and transparent data sharing practices.

Experimental design

See section 1

Validity of the findings

See section 1

Additional comments

See section 1

Reviewer 2 ·

Basic reporting

I invite you to improve the introduction, the background where you demonstrate how the work fits into the wider field of knowledge.
There are several bibliographic citations found in the body of the paper but not reflected in the bibliographic reference e.g. Ruth et al., 2008 and 297 Biddle, A., Stewart, L., Blanchard, J., Leschine, S. (2013).Untangling the genetic basis of fibrolytic specialization by Lachnospiraceae and Ruminococcaceae in diverse gut communities [J]. Diversity, 5(3): 627-640. Literature used in the article should be properly cited.

Experimental design

Despite the undoubted merits of the paper, there are some weaknesses in the experimental design that, in my opinion, preclude its publication in a high-quality scientific journal.
The most serious concern is the poor characterisation of the experimental groups. The composition and mode of food use are unclear.
As for sample collection and sequencing analysis, although some references are given, this is a key point of the article and these procedures should be minimally described to allow readers to better understand the nature of the treatments tested. In addition, the characterisation of the diets is poor. It is difficult to be sure that the differences found between treatments are due to differences between the treatments used and not to uncontrolled variability in the procedures employed.
The study also lacks a basic analysis of the intestinal environment of the pigs to support the discussion (pH of digestion, basic microbiology, etc.). It is not made clear what was considered as experimental unit. The title of the tables in this respect is rather confusing.

Validity of the findings

There is a lot of inconsistency in the discussion, there are many gaps and confusion, there are many gaps that do not allow to see the novelty in the specific audience. They do not use the consideration of current studies which does not allow to describe in a clear and coherent way, which does not allow to replicate the study.
The deficiency of the information provided in the materials and methods section, as well as the results, does not allow you to come to a proper formulation of the conclusion, where you can answer the question with solid results of the study.

Additional comments

I invite the authors to profoundly improve the form and substance of the document, because there are many gaps that call into question the valuable findings reported in this study.
In addition, I suggest that the bibliographic citation throughout the document be revised, because there is bibliography that is found in the body of the document and does not appear in the references, or vice versa. Because the way it is it looks like a first draft.

Annotated reviews are not available for download in order to protect the identity of reviewers who chose to remain anonymous.

---

## Round 0.2 · Major Revisions

Two expert reviewers have evaluated your resubmitted manuscript. Both comment on the major improvements that have been made to the manuscript. However, there are additional comments that have been made that I agree should be taken into account in a revised version.

Please ensure that all of the points that were discussed in your rebuttal letters are incorporated into the new version of the manuscript. Please clearly indciate what modifications have been made and where they are incorporated.

Reviewer 1 ·

Basic reporting

I would like to sincerely thank the authors for addressing all my revision requests. Additionally, I appreciate that the authors included in the bibliography of their manuscript the studies suggested during the first round of revisions. Compared to the initial submission, the manuscript has significantly improved in terms of both presentation and, most notably, content.

Although the authors responded comprehensively to all my requests in the rebuttal letter, I noticed that the table reporting the growth performance results of the pigs has not been included in the revised manuscript, and it should be added.

In the rebuttal letter, the authors have provided excellent responses to all the revision requests. However, everything discussed in the rebuttal letter must be incorporated into and explicitly stated in the manuscript.

Moreover, I have a suggestion for the authors. Based on their interesting results, it would be appropriate to perform a differential analysis using the Rstudio package called DESeq2. This package allows for differential analysis at the genus level, aiming to identify which genera are differentially expressed across the three diets. By referring to the Log2FoldChange values, it is possible to determine which genera are upregulated or downregulated in the different dietary groups. Considering that the pigs were fed a basal diet (control) and 5%, 10%, and 15% WPCS, the following comparisons could be evaluated in the differential analysis: control vs 5%, control vs 10%, and control vs 15%. This approach would help assess which bacterial genera are differentially expressed, that is, positively or negatively modulated in the three dietary groups. If the authors obtain any interesting results, it would be advisable to integrate them into the results section and subsequently into the discussion. If necessary, the authors can use the platform https://www.microbiomeanalyst.ca/ to perform the DESeq2 analysis more efficiently. I look forward to their feedback on this matter.

Experimental design

In the 'Experimental Design' section, please include the controlled ambient temperature and relative humidity of the pig housing facility.

Validity of the findings

Please, see other sections.

Additional comments

- Line 60: In the abstract, "Simpson" should be capitalized. The same correction is needed on line 803.
- Line 70: Correct to "Picrust".
- Line 73: "Singal" should be corrected to "signal". The same applies to line 1047.
- Minor typographical errors in the text can be addressed during the proof-correction stage.
- In the 'Experimental Design' section, please include the controlled ambient temperature and relative humidity of the pig housing facility.

Reviewer 2 ·

Basic reporting

I suggest that the language be improved, however, the final document is better than the first one and I agree that it should be published.

Experimental design

I suggest that the language be improved, however, the final document is better than the first one and I agree that it should be published.

Validity of the findings

I suggest that the language be improved, however, the final document is better than the first one and I agree that it should be published.

---

## Round 0.3 · Minor Revisions

I am satisfied with the technical and scientific modifications that have been made to the manuscript. However, there are still quite a few errors in the use of the English language throughout the manuscript which need to be corrected to be acceptable for publication in PeerJ. For example in the Background section of the Abstract: In the first sentence "sillage" is singular therefore the verb should be "has" not "have". The last sentence of the background does not make sense; the word "While" should be removed.

---

## Round 0.4 · Minor Revisions

Please correct the following in the manuscript:

Line 68 delete “in”
Line 73 delete “one of”
Line 75 what is acidic – odour?
Line 144 to kept should read and kept
Line 149 of pig should read of the pig
Line 200 analysis should read analyses
Line 206 different addition level should read different levels
Line 208 gain refers to gain of what? Weight?
Line 248 was should read are
Line 248 using volcano should read using a volcano
Line 250 Delete The
Line 254 experiment should read experimental
Line 253 to 255 Sentence does not make sense. Please correct. What parameter was observed to significantly decrease?
Line 255 to 258 Sentence does not make sense. Please correct. What parameter had higher relative abundance?
Line 257 comparing should read compared
Line 384 fiber should read fibers
Line 391 delete significant
Line 392 exhibiting high should read exhibiting significantly high
Line 392 swine intestinal what? Or should it read swine intestines? (intestinal is an adjective)
Line 299 This probably should read This was probably
Line 409 healthy should read health

Please ensure that the final version uses correct English throughout the manuscript.

---

## Round 0.5 · accepted · Accept

Thank you for making the suggested changes to the manuscript. I now find it suitable for acceptance by PeerJ.